# The impact of tinnitus on Dutch general practices: A retrospective study using routine healthcare data

Julia M. Bes[1]*, Robert A. Verheij[1,2,3], Bart J. Knottnerus[1], Karin Hek[1]

1 Nivel, Netherlands Institute for Health Services Research, Utrecht, The Netherlands, 2 Tilburg School of Social and Behavioral Sciences TSB: Tranzo, Tilburg University, Tilburg, The Netherlands, 3 The National Health Care Institute, Diemen, The Netherlands

* j.bes@nivel.nl

## Abstract

### Purpose

Global prevalence of tinnitus (15%) is rising, indicating an increase in patients seeking help for this common symptom and potentially affecting the accessibility of care. The aim of this retrospective study is twofold; describing the morbidity trends and healthcare utilization among patients with recorded tinnitus at Dutch general practices (GP), and comparing overall healthcare utilization before and after tinnitus to similar patients without recorded tinnitus.

### Patients and methods

Routine electronic health records data from general practices participating in Nivel Primary Care Database were used to describe trends in age- and sex-specific incidence, contact prevalence and healthcare utilization (contacts, prescriptions, and referrals to secondary care) for tinnitus from 2012 to 2021. Patients with tinnitus were matched (on sex, age, general practice) to patients without tinnitus (1:2). Healthcare use one year before and after a first GP contact for tinnitus (period) was analyzed with negative binominal (number of contacts) and logistic regression (prescriptions or referrals).

### Results

Between 2012 and 2021, tinnitus incidence increased by 33% (3.3 to 4.4 per 1000 person-years). The largest increase was observed among 20-44-years (2.6 to 3.8 per 1000 person-years). One year after the first GP contact for tinnitus, patients with tinnitus utilized healthcare more frequently than patients without tinnitus. The referral rate increased the most (OR 1.99, 95%CI 1.83–2.16). The year before tinnitus at the GP, patients with tinnitus had more often GP consultations (IRR 1.16, 95%CI 1.12–1.19) and referrals (OR 1.09, 95%CI 1.01–1.18) than patients without tinnitus.

**Data Availability Statement:** Data of the Nivel-PCD contains potentially identifying or sensitive patient information. Access to data in the Nivel Primary Care Database (Nivel-PCD) is subject to Nivel Primary Care Database governance codes.

Requests for access to the data can be directed at gegevensaanvragen@nivel.nl Restrictions involve approval by the appropriate Nivel Primary Care Database governance bodies (privacy committee and steering committee).

**Funding:** Nivel-PCD is funded by the Dutch Ministry of Health, Welfare and Sports.

**Competing interests:** The authors have declared that no competing interests exist.

## Conclusion

Although GPs saw an increased number of patients with tinnitus, absolute numbers of patients remained low. Patients' healthcare use increased after the first tinnitus complaint at the GP, mainly for secondary care. Yet, they already used healthcare services more frequently than similar patients without tinnitus. Insight into possible explanations for these health disparities could contribute to targeted prevention.

## Introduction

Tinnitus, characterized by hearing sounds without an acoustic source being present, is a common health problem. The experienced severity can range from mild ringing without bother to a loud and persistent noise that can affect sleep and significantly reduce quality of life [1–3]. Worldwide, an estimated 740 million adults experience any tinnitus (overall prevalence ranging ranged from 5% and to 43% of the adult population, depending on severity), suggesting that the global burden is large [4, 5]. Multiple risk factors are associated with tinnitus, such as loud noise exposure (e.g, during leisure time activities) and hearing loss [6, 7]. Two large cohort studies in Wisconsin (USA) suggest that the prevalence of tinnitus is increasing over time (1.4% in each 5-year birth cohort) [8, 9].

With an increase in prevalence, healthcare providers may see an increased number of patients experiencing tinnitus [9]. Various treatments and care for patients with tinnitus are available, mainly to treat its comorbidities [10]. There is no specific drug proven effective against tinnitus available [11]. With no curative treatment to date, this can lead to substantial healthcare costs and a high burden of disease for society [12–14] A Korean study with medical claims data (2010–2018) showed that the number of patients, medical utilization, and expenditures for tinnitus steadily increased during the studied period [15]. The increased demand for care may be first apparent to general practitioners in healthcare systems where they are the first point of contact for care, such as the Netherlands. This could potentially put accessibility and quality of care for themselves and others under pressure. However, longitudinal studies about the healthcare utilization of patients with tinnitus are scarce.

Comparison in healthcare utilization between patients with and without tinnitus could put findings in perspective. A cross-sectional study by Rademaker et al. indicated that healthcare use is higher in patients with tinnitus compared to patients without. For example, patients with tinnitus had in total more consultations in primary care than people without tinnitus (9.8 (sd 10.9) vs 5.7 (sd 7.9)), independent of age and sex [16]. However, this cross-sectional comparison could not establish causality. Longitudinal studies can contribute to unraveling whether this is attributed to tinnitus or whether this was present before the development of tinnitus. Given the preventable nature of some risk factors for tinnitus (e.g., loud noise exposure), a substantial increase in healthcare utilization due to tinnitus could add to the need for public health programs to reduce risk.

Therefore, our study aim is twofold. First, we describe developments in age- and sex-specific tinnitus morbidity and healthcare utilization at the GP in the past decade. Second, we compare healthcare use at the GP of patients with and without tinnitus before and after their initial GP contact for tinnitus. From a health policy point of view, insight into healthcare utilization and management for and of patients with tinnitus could inform healthcare resource allocation.

## Material and methods

### Data source

This retrospective study was performed using data from electronic health records from general practices participating in Nivel Primary Care Database (Nivel-PCD). Nivel-PCD is a nationally representative longitudinal database that includes routinely recorded data from electronic health records (EHR) from 10% of the general practices in the Netherlands. Pseudonymized patient-level data contains data on demographics, GP contacts, diagnoses, prescriptions, and referrals to secondary care. Complaints and diagnoses at the GP are recorded using the International Classification of Primary Care version 1 (ICPC-1) [17]. Prescriptions are coded using the Anatomical Therapeutic Chemical Classification system (ATC) [18].

Nivel-PCD contains data that is pseudonymized before leaving the healthcare organization's premises and therefore does not comprise any directly identifying personal information such as names, addresses and citizen service number. According to Dutch law (Dutch Civil Law, Article 7:458), neither obtaining informed consent from patients nor approval of a medical ethics committee is obligatory for this type of observational study containing no directly identifiable data. This study was approved by the applicable governance bodies of Nivel-PCD under number NZR-00323.004 (dd 06-02-2023).

### Study design and sample selection

For this study, we established two separate retrospective cohorts. First, in our descriptive cohort, morbidity trends and healthcare utilization of patients with recorded tinnitus were described between 2012 and 2021. Practices were included whose EHR data fulfilled data quality criteria per type of healthcare utilization (i.e. contacts, prescriptions, and referrals) [19]. Patients were selected if they had a recorded disease episode of tinnitus (ICPC H03) [19]. Data on referrals was available from 2015 onwards in Nivel-PCD. Second, in our matched cohort, overall GP care utilization was compared between patients with and without tinnitus, between 2015 and 2021. Practices were included who fulfilled data quality criteria as described above [19]. Practices were selected if they had data available for four consecutive years between 2015 and 2021. Patients were regarded as incident cases if they had a GP contact with a recorded ICPC code for tinnitus (H03) in Nivel-PCD [19]. Within Nivel-PCD, first contact is defined as the start of a constructed episode of illness (i.e., episode of care, consultation, or prescription) for that specific disease, calculated as described in Nielen et al. [19] Patients were included if they had been registered as patients in a practice for at least two years before the first GP contact known for tinnitus (i.e. to prevent misclassification of cases and controls) [20], the year of initial tinnitus contact, and one year after the first contact for tinnitus. The initial GP contact for tinnitus known in the Nivel-PCD was regarded as the index date. Controls, who were selected from the general population within Nivel-PCD, had data available for at least four consecutive years, had at least one GP contact in the year on which they can be matched to cases (i.e. to reduce heterogeneity between cases and controls), and had not any contact for tinnitus during the entire period recorded in the Nivel-PCD. Matching of cases and controls was done 1:2 by age group (five years), sex, general practice, and year in which they could be matched. Controls received the same index date as their matched case. To compare healthcare utilization between cases and controls, we selected contacts, prescriptions, and referrals to secondary care one year (365 days) before and one year (365 days) after the index date.

## Measures

For the descriptive cohort, yearly incidence and contact prevalence rates of tinnitus were calculated [21]. Incidence was calculated by dividing the number of new episodes for tinnitus by the total number of person-years in each year. Contact prevalence was calculated as the number of patients with at least one GP contact for tinnitus divided by the total number of person-years that year. These measures were standardized to the Dutch population per year on age (five-year groups), sex, and degree of urbanization of the practice. Sex was specified as male/female. Age was divided into five categories (0–19 years, 20–44 years, 45–64 years, 65–74 years, 75 years and older). The degree of urbanization was divided into three categories (strongly to extremely urbanized, areas with more than 1,500 addresses/km2; moderately urbanized, areas with 1,000 to 1,500 addresses/km2; hardly to not urbanized, areas with less than 1,000 addresses/km2) [22]. To calculate the yearly prescription rate for tinnitus, the number of patients with at least one prescription where an ICPC code for tinnitus has been recorded after GP contact was divided by the number of patients with a contact for tinnitus. The annual referral rate for tinnitus, the number of tinnitus patients with at least one referral to secondary care for tinnitus was divided by the number of patients with a contact for tinnitus.

For the matched cohort, demographic variables (i.e. sex, mean age, degree of urbanization of GP) and number of chronic diseases before the index date were calculated. In Nivel-PCD, a list of 109 chronic diseases was compiled, where chronic is defined as an episode of illness for that specific disease that terminates only when a patient dies [19]. The presence of chronic diseases at index date was dichotomized into two categories (zero and one or more). For the comparison of healthcare utilization between cases and controls in the period before and during tinnitus, three measures were included; total number of contacts with the GP, percentage of patients with a prescription, or percentage of patients with a referral to secondary care. For cases and controls, healthcare use on the index date was not included in the before or after period counts to avoid overestimation in one of the periods.

## Statistical analysis

For the descriptive cohort, baseline characteristics of GPs and patients are presented. Continuous variables (e.g., age) are presented with mean and standard deviation (sd). Dichotomous and categorical variables (e.g., sex) with numbers and percentages. Standardized yearly incidence and contact prevalence are presented per 1,000 person-years, stratified by sex and age categories. Yearly percentage of patients that received a prescription or referral after contact for tinnitus is presented stratified by sex. In addition, the top five most prescribed medications at the ATC2 level (i.e. N05, N06, N07, R01, S02) and top three most referred to specialism (i.e. ENT-specialist, Neurology, Psychiatry) for patients with tinnitus are presented, stratified by age categories. For the matched cohort, descriptive statistics are presented to compare the characteristics of cases and controls. Continuous variables (i.e., age, number of contacts) are presented with mean and sd. Categorical variables (i.e., sex, urbanization, patients with a prescription or referral) are presented with numbers and percentages. To analyze the healthcare utilization categories, the number of GP contacts was compared between cases and controls with a negative binomial regression. Binary variables (i.e., prescription and referral) were compared with a logistic regression. Because not all participating practices in the Nivel-PCD have available data on referrals, a selection of cases and controls was used for this analysis. To determine if overall healthcare utilization significantly differed between before and during tinnitus, an interaction term was added in all analyses. Standard errors are corrected for intragroup

correlation per GP practice. A p-value <0.05 was held significant. All analyses were done with StataSE version 18 [23].

## Results

### Morbidity of and healthcare utilization for tinnitus

Baseline characteristics of the descriptive cohort are presented in S1 Table. Between 2012 and 2021, tinnitus incidence increased by 33% (from 3.3 to 4.4 per 1,000 person-years). The tinnitus incidence was higher among men than among women, but trends over time were similar (Fig 1). For age categories, the tinnitus incidence was highest for patients aged 45 years and older, but the largest increase was observed among 20- to 44-years (2.6 to 3.8 per 1,000 person-years (S1 Fig)). Contact prevalence (i.e. the number of patients with a GP contact) for tinnitus was slightly higher than incidence but increased similarly (Fig 1 and S2 Fig).

About 20% of the individuals who contacted their GP for tinnitus received a prescription for tinnitus each year. This remained stable between 2012 and 2021 (Fig 2). Of these 20%, nasal corticosteroids were prescribed the most. Per increasing age category, the percentage of tinnitus patients with nasal corticoids decreased and the percentage of patients receiving other medications, such as antidepressants, increased (S3 Fig). Approximately 20% of tinnitus patients were referred to a medical specialist each year for tinnitus after one or more contacts with their GP. This fluctuated around 20% to 25% per year and was mainly to the ENT-specialist (S4 Fig).

### Healthcare utilization of patients with and without tinnitus

In total, 9,097 patients with tinnitus had data available between 2015 and 2021 two years before and one year after their first GP contact for tinnitus. These patients were matched to two comparable controls without tinnitus known at the GP (N = 18,158), as seen in Table 1. Mean age was 53 years and 47% was female. The number of chronic diseases before tinnitus was not

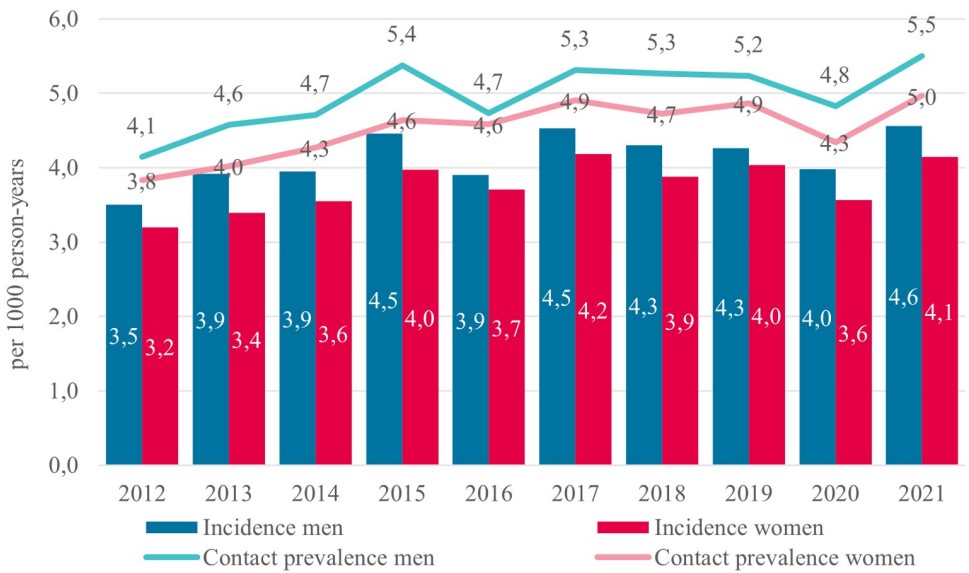

**Fig 1. Standardized yearly incidence and contact prevalence of tinnitus in Dutch general practices between 2012 and 2021, stratified by sex.** Incidence: the number of new tinnitus cases divided by total person-years. Contact prevalence: the number of patients with a GP contact for tinnitus divided by total person-years.

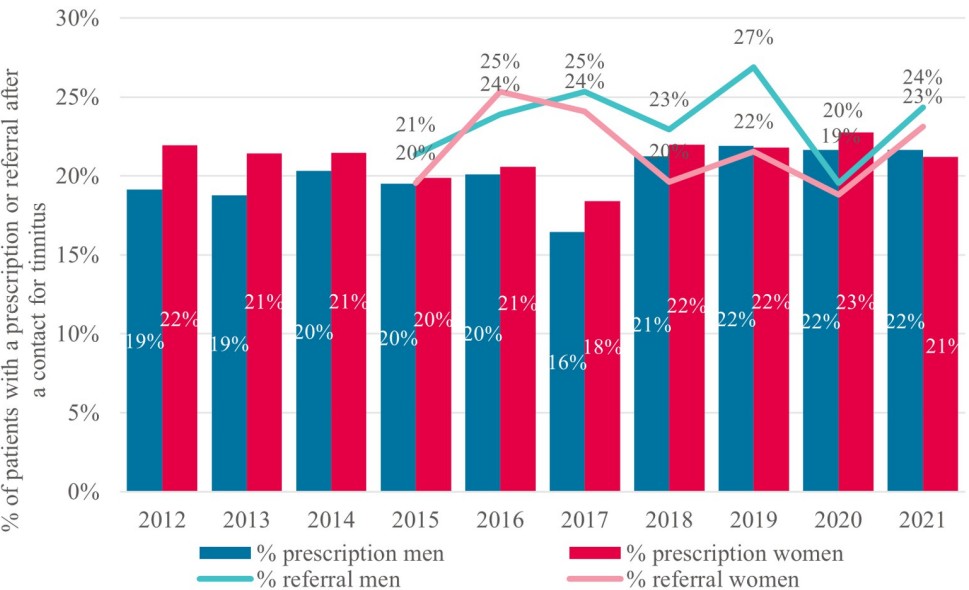

**Fig 2. Yearly proportion of patients receiving a prescription or referral for tinnitus between 2012 and 2021, stratified by sex.**

used as a matching variable but was similar between groups. One year before and after the index date, patients with tinnitus contacted their GP more often compared to similar patients without a known tinnitus. The percentages of patients with a prescription or referral were also higher in patients with tinnitus compared to patients without.

Patients with tinnitus (cases) consulted their GP more often than similar patients without tinnitus (controls), before (IRR 1.7, 95%CI 1.1–1.2) as well as after (IRR 1.3, 95%CI 1.3–1.3)

**Table 1. Descriptive statistics of the matched cohort; patients with a first contact for tinnitus at the GP matched (on age, sex, and practice) to patients without known tinnitus.**

| | Controls (n = 18,158) | Cases (n = 9,079) | p-value (p<0.05) |
|---|---|---|---|
| **Age, mean (sd)** | 52.9 (17.4) | 53.0 (17.3) | 0.92 |
| **Female, n %** | 8,602 (47.4%) | 4,301 (47.4%) | 1.00 |
| **Urbanization of general practice** | | | 1.00 |
| Low | 5,388 (29.7%) | 2,694 (29.7%) | |
| Medium | 4,490 (24.7%) | 2,245 (24.7%) | |
| High | 8,280 (45.6%) | 4,140 (45.6%) | |
| **Chronic conditions before tinnitus, n (%)** | | | |
| No chronic disease | 5,442 (30,0%) | 2,661 (29,3%) | |
| One or more chronic disease(s) | 12,716 (70,0%) | 6,418 (70,7%) | 0.26 |
| **Number of GP contacts (one year before and after index date), mean (sd)** | 11.2 (11.2) | 14.7 (13.2) | <0.001 |
| **Patient with prescription (one year before and after index date), n (%)** | 16,612 (91.5%) | 8,449 (93.1%) | <0.001 |
| | **Controls (n = 7,298)** | **Cases (n = 3,649)** | |
| **Patient with a referral (one year before and after index date), n (%)[a]** | 3,256 (44.6%) | 2,398 (65.7%) | <0.001 |

sd = standard deviation.

[a]Referral data was only available for a selection of general practices

**Table 2. Predicted margins and standard error of overall healthcare use between cases and controls, one year before and after the index date.**

|  | Period | Controls | Cases | IRR (95% CI) |
|---|---|---|---|---|
| **Mean number of contact(s), SE** | Before | 5.5 (0.07) | 6.4 (0.10) | 1.16 (1.12–1.19)* |
|  | After | 5.7 (0.08) | 7.3 (0.11) | 1.29 (1.26–1.32)* |
|  |  |  |  | OR (95% CI) |
| **% Patients with prescription(s), SE** | Before | 83.8 (0.42) | 82.5 (0.48) | 0.98 (0.92–1.04) |
|  | After | 83.1 (0.46) | 85.2 (0.48) | 1.17 (1.09–1.25)* |
| **% Patient with referral(s), SE** | Before | 25.4 (0.83) | 27.2 (0.90) | 1.09 (1.01–1.18)* |
|  | After | 28.5 (0.61) | 44.2 (0.96) | 1.99 (1.83–2.16)* |

SE = standard error. IRR = incidence rate ratio. OR = odds ratio. CI = confidence interval.
*statistically significant ($p < 0.05$)

first contact for tinnitus (Table 2). For the percentage of patients with a prescription, there is no significant difference between cases and controls before tinnitus. Comparing one year during tinnitus with before, the percentage of controls receiving a prescription remains constant (OR 1.0, 95%CI 1.0–1.1) while cases have a 1.2 times higher odds (95%CI 1.1–1.3) of having one or more prescriptions. Patients with tinnitus are significantly more referred to secondary care compared to similar controls, both before (OR 1.1, 95%CI 1.0–1.2) and after (OR 2.0, 95% CI 1.8–2.2) first contact for tinnitus. Comparing the year after first contact for tinnitus with before, controls and cases have respectively a 1.2 (95%CI 1.1–1.3) and 2.1 (95%CI 1.9–2.4) times higher odds of having a referral.

## Discussion

This is the first study that assesses the morbidity and healthcare utilization of patients with tinnitus in Dutch general practice longitudinally. Our findings reveal that the number of patients contacting their GP for tinnitus is low but has increased in the past decade, especially among those aged 20- to 44-years. One in five patients with tinnitus received a prescription or referral for tinnitus. Comparing overall healthcare utilization showed an increased odds ratio for all types of healthcare use for patients after their first contact for tinnitus compared to similar patients without tinnitus, mainly for secondary care. However, looking at one year before tinnitus was known at the GP, we found that patients with tinnitus already had more often GP consultations and were more often referred to secondary care than similar patients without known tinnitus.

### Comparison with existing literature

Prior research explored the incidence and perceived severity of self-reported tinnitus within the general population, but lacked an examination of the healthcare perspective on the impact and burden of tinnitus. Survey studies have indicated that the number of people suffering from tinnitus is larger than found in the current study [4, 5]. Surveys often make use of self-reported tinnitus, where definitions can differ, but contain more detailed information about e.g. duration and severity [4, 5, 9]. Individuals might not visit their GP for less severe tinnitus, even though they can indicate by questionnaire that they suffer from this complaint. A survey study in the US showed that around half of the respondents who experienced tinnitus in the last year discussed this with their doctor [24]. A study in the United Kingdom (UK) suggested that tinnitus of sufficient concern to warrant seeking medical attention could be defined as clinically significant tinnitus, with an annual rise in IR of 2.1 cases per 10,000 person-years

between 2002 and 2011 [25]. A study using the UK Clinical Practice Research Datalink estimated on average 25.0 new tinnitus cases per 10,000 person-years (95% CI: 24.6–25.5) between 2000 and 2016 [20]. These results are comparable to our study findings. Although no consensus on incidence exists, the consistent increase of tinnitus cases at GPs over time emphasizes a rising health concern. While incidence in the general population is estimated to be much higher, the rise could suggest that tinnitus is having an increased effect on public health and society.

Our finding that patients at older age present themselves more with tinnitus is similar to the literature [4, 20, 25]. Although the underlying causes of tinnitus are not yet fully understood, age is one of its known risk factors. A literature review revealed that at older age chronic tinnitus is subjectively louder, more annoying, and more distressing than in younger patients [26]. However, we found the highest increase in tinnitus incidence of 20-44-year-old patients in our study. It is possible that the observed increase in tinnitus prevalence by Nondahl et al. in 2002 (1.4% per 5-year birth cohort), persisted or worsened among today's adolescents and older individuals [9]. In addition, exposure to loud noise during leisure time activities such as visiting concerts and nightclubs as well as mobile technology usage (e.g. cell phones and music players) is a risk factor for hearing loss or tinnitus, and may be of particular importance to young people [27–29]. Another possible explanation for the increase in prevalence might be the level of awareness of tinnitus, for example, due to media attention for tinnitus that raise awareness in the general population. This might increase help seeking behavior at GPs [30]. Regardless of the reasons, this study adds to the understanding of the impact of tinnitus on different populations and can help tailor care and interventions based on specific demographic characteristics.

Given that the majority of European countries do not have national clinical guidelines for tinnitus, management and treatment can vary widely across countries and GPs [31]. In the Netherlands, there is no guideline specific for tinnitus that guides GPs. Hence, approaches for management differ [32–35]. Literature reports little about the magnitude of patients receiving medication for tinnitus [36]. This might be explained by the fact that pharmaceutical treatment for tinnitus is not recommended since there is no evidence for effectiveness in reducing tinnitus complaints. In addition, a meta-analysis indicated that none of the investigated pharmacological interventions was associated with changes in quality of life compared to placebo/control [10, 37]. Still, we found that GPs prescribed medication for 20% of the patients with tinnitus. A possible explanation for this might be the heterogeneity in causes of tinnitus. Nasal corticosteroids are used for different diseases mostly rhinosinusitis (related) problems, which impact normal hearings and may cause the symptoms of tinnitus [38]. Per increasing age category, our study found that more often psycholeptics and psychoanaleptics were prescribed for tinnitus. Although a systematic review indicated that these medications may have effects on symptom relief in patients with tinnitus [39], they are not part of clinical guidelines on tinnitus treatment due to limited evidence of their direct effectiveness [10, 11]. Since a variety of medications for tinnitus are prescribed in Dutch GPs, this may underline that a GP guideline for tinnitus treatment per specific age category could contribute to more uniform prescribing behavior.

After a first contact for tinnitus, the increase in all types of overall healthcare use indicates that tinnitus itself is contributing to increased healthcare utilization which is comparable to other studies [16, 40]. Our study showed that referral rate differed the most between patients with and without tinnitus. Referral to an ENT specialist may be recommended based on the duration and type of tinnitus, to rule out any serious or treatable medical condition [11, 34]. In the Netherlands, the ENT specialist is in the top five specialists referred to by the GP and when zooming in on specific diagnoses, tinnitus is in the top three [41]. The high referral rate for

tinnitus by GP might be explained by the fact that there is a tinnitus guideline for medical specialists in the Netherlands. This guideline proposes several audiological and diagnostic examinations and various treatment options to reduce tinnitus complaints [42]. A review by Carmody et al. indicated that the majority of help-seekers for tinnitus expressed dissatisfaction and reported negative interactions with healthcare providers, except in specialized tinnitus clinical settings [43]. The steady increase of patients presenting with tinnitus to their GP may indicate a rise in the absolute number of patients requiring secondary care. Depending on the number of consultations they require with a specialist, this could result in increased healthcare use in secondary care and substantial costs in the future.

Interestingly, our study reveals that one year before the first GP contact for tinnitus, these patients already seek more medical help in general. This is evident from the higher number of contacts with the GP and referral rate compared to their matched controls. The higher referral rate before tinnitus was known at the GP could imply that patients with tinnitus have already unmet medical needs that cannot be resolved in primary care alone. Factors associated with the presence of tinnitus have mainly been studied cross-sectionally [6, 7, 9, 16], whereby underlying causes for the development of tinnitus are relatively unknown. One longitudinal study revealed that risk factors for new-onset tinnitus five years later were a history of smoking (OR 1.5, 95%CI 1.0–2.2) and higher levels of somatization (OR 2.0, 95%CI 1.2–3.3) [44]. The latter is defined as "a tendency to experience and communicate somatic distress in response to psychosocial stress and to seek medical help for it" [45]. Although psychological problems such as anxiety or insomnia are known as common comorbidities after the development of tinnitus [46, 47], our result could suggest that these complaints may already have been present before tinnitus. Although underlying causes of higher healthcare use before tinnitus were outside the scope of this study, further research could help unravel risk factors for targeted prevention.

In context, the prevention of (aggravation of) tinnitus and related healthcare use could contribute to the quality of life of patients and reduce the economic burden. It is well-known that quality of life is compromised by tinnitus due to insomnia, emotional and cognitive distress, and depression [1–3, 46]. Multiple studies have investigated the cost of tinnitus regarding individuals' health and well-being, health care utilization, and society, with societal costs estimated as the highest [12–14, 32, 48, 49]. For example, Tziridis et al. estimated that per person with tinnitus the cost of economic loss due to sick leave (EUR 2302) exceeds the cost that accounts for public healthcare (EUR 2207) [49]. Debatably, implementing a proactive and preventive approach may result in patients with no or less severe tinnitus and lower costs in primary and secondary care as well as for society. For example, using hearing protection in places with loud noise exposure [50]. It should be investigated if these approaches could be cost-effective and significantly reduce the burden of disease, thereby contributing to the future affordability and accessibility of healthcare.

## Strengths and limitations

This study's longitudinal approach contributes valuable insights into the evolving healthcare utilization of patients with tinnitus over time. When interpreting the results, limitations should be taken into account. Firstly, using routine healthcare data for tinnitus we could have underestimated the estimates in GP care. Tinnitus is a condition that cannot be confirmed with objective diagnostic procedures by the GP. It could be likely that GPs record cases of tinnitus as 'other ear complaints' (i.e., hearing impairment). Hence, tinnitus reported by GPs does not encompass all cases, and actual prevalence rates are likely to exceed those outlined in our research. Second, referrals included in our study are to secondary care only, while referrals to

for example an audiological center that could be made for tinnitus are not included here. This could indicate that the referral rate for tinnitus by GP is even higher and that healthcare utilization in other healthcare settings could increase as well. Nevertheless, the utilization of the Nivel-PCD, a large routine healthcare database that includes 10% of Dutch GPs and the standardization to the Dutch population, enhances the reliability of the estimated results for GP care. Third, although controls were matched on several factors, no adjustment was made for possible other confounders. However, matching to comparable controls puts our findings into perspective. Misclassification of controls is avoided as much as possible by using multiple years and cases and controls are matched on multiple demographic factors, thereby enhancing the validity of the study results.

## Implications of study results

While only a small proportion of patients contact their GP for tinnitus, up to one in five of them receive a prescription or a referral to secondary care for tinnitus. This could indicate that a large proportion of these patients require medical attention other than from their GP. This finding underscores the importance of understanding and addressing the healthcare needs of tinnitus patients beyond primary care, while also acknowledging that GPs may be constrained by facilities or expertise, which could affect their ability to provide appropriate care for these patients. Furthermore, the observation that before the first contact for tinnitus patients already utilized healthcare services more often than comparable patients without tinnitus highlights existing health disparities. Insight into the explanatory factors for this difference can highlight the specific needs of this patient population and can lead to more targeted prevention or support services in the future.

## Conclusions

To conclude, this study provides a comprehensive understanding of the healthcare utilization of patients with tinnitus in Dutch general practice. GPs saw an increased number of patients with tinnitus in the past decade, yet absolute numbers remain low. Healthcare utilization increased after the first known GP contact for tinnitus, mainly for secondary care, but patients with tinnitus utilized healthcare already more often than similar patients without tinnitus. From a health policy point of view, insight into possible explanations for these health disparities needs to be further researched for targeted prevention.

## Supporting information

**S1 Table. Baseline characteristics of the descriptive cohort population.** GP = general practice. sd = standard deviation.
(TIF)

**S1 Fig. Standardized yearly incidence of tinnitus, stratified per age category.** Incidence: the number of new diagnoses divided by total person-years, per 1000 person-years.
(TIF)

**S2 Fig. Standardized yearly contact prevalence of tinnitus, stratified per age category.** Contact prevalence: the number of patients with a general practice contact for tinnitus divided by total person-years, per 1000 person-years.
(TIF)

**S3 Fig. Type of prescription for tinnitus after a general practice contact for tinnitus, stratified by age category.**
(TIF)

**S4 Fig. Type of referred specialism for tinnitus after a general practice contact for tinnitus, stratified by age category.** ENT = ear, nose and throat.
(TIF)

## Author Contributions

**Conceptualization:** Julia M. Bes, Robert A. Verheij, Karin Hek.

**Formal analysis:** Julia M. Bes.

**Methodology:** Julia M. Bes, Karin Hek.

**Supervision:** Robert A. Verheij, Karin Hek.

**Validation:** Julia M. Bes, Karin Hek.

**Writing – original draft:** Julia M. Bes, Karin Hek.

**Writing – review & editing:** Robert A. Verheij, Bart J. Knottnerus, Karin Hek.

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
