## [Decision Letter · Decision Letter 0]

9 Jul 2024

PONE-D-24-13714Tinnitus; a growing public health problem, but what is the impact on Dutch general practices? A retrospective study using routine healthcare dataPLOS ONE

Dear Dr. Bes,

Thank you for submitting your manuscript to PLOS ONE. After careful consideration, we feel that it has merit but does not fully meet PLOS ONE’s publication criteria as it currently stands. Therefore, we invite you to submit a revised version of the manuscript that addresses the points raised during the review process.

We look forward to receiving your revised manuscript.

Kind regards,

Sarah Michiels

Academic Editor

PLOS ONE

Reviewers' comments:

Reviewer's Responses to Questions

**Comments to the Author**

1. Is the manuscript technically sound, and do the data support the conclusions?

Reviewer #1: Yes

Reviewer #2: Yes

2. Has the statistical analysis been performed appropriately and rigorously? 

Reviewer #1: Yes

Reviewer #2: Yes

3. Have the authors made all data underlying the findings in their manuscript fully available?

Reviewer #1: Yes

Reviewer #2: Yes

4. Is the manuscript presented in an intelligible fashion and written in standard English?

Reviewer #1: Yes

Reviewer #2: Yes

5. Review Comments to the Author

Reviewer #1: PLOS ONE Tinnitus; a growing public health problem, but what is the impact on Dutch general practices? A retrospective study using routine healthcare data

In this manuscript the authors describe an interesting retrospective study to report the morbidity trends and healthcare utilization among patients with tinnitus at Dutch general practices (GP), and comparing overall healthcare utilization before and after tinnitus to similar patients without tinnitus using health care databases. It is a topic of high interest, with clear methods, high quality data and adequate description of results and its discussion. I strongly recommend studies on this for providing knowledge on health care consumption and its potential impact on applying health care resources to those in need. However, the manuscript misses quality based on flaws in details on tinnitus and rationale for statements quoted throughout the manuscript. I strongly advise to revise the manuscript on this which at this moment strongly limit the quality of the report and could easily be adjusted.

Introduction

Line 45; this is the definition of subjective tinnitus; please add this to the sentence.

Line 48: ‘ranging from 5% and 43%’; details are missing about these percentages, for example ‘of people from the adult general population’?. Besides this, it’s incorrect to state that with respect to these numbers they all ‘suffer from tinnitus’. About at least 25% of the people with tinnitus do not have any complaints from their tinnitus. Please be cautious with this phrasing on suffering (e.g. also in line 228)

Line 51; ‘by occupation and mobile technology usage, its prevalence is expected to rise (1.4% in each 5-year birth cohort). ‘ � this statement is speculative. So far there is no robust evidence on the rise of the prevalence nor on the etiology of this ‘expected rise’. So please refrain from speculations and describe this as an hypothesis, knowledge gap or topic of interest for future health care purposes. This is the same reason that also the title of the manuscript is somewhat misleading: ‘Tinnitus; a growing health problem….’. As the evidence base for this statement is absent and the current manuscript is on GP diagnosis we can’t make firm conclusions on this. Therefore I would suggest to rephrase this title slightly

Line 55: ‘For example, drug therapies used to treat tinnitus, like antihistamines, 56 antidepressants, or antiepileptics, may provide symptom relief but do not offer a complete cure for tinnitus. � this statement is not according to national treatment guidelines nor according to the current evidence base on the effectivity of the mentioned therapies.

Line 65; ‘cross-sectional study by Rademaker et al. indicated that healthcare use is higher in 66 patients with tinnitus.’ please indicate the comparison which was made (in this case compared to those without a diagnosis of tinnitus).

Line 96: ‘patients with tinnitus’; Although in the text it is mentioned that ICPC-diagnosis were obtained, from this, the reader can’t understand that if you refer to ‘patients with tinnitus’ you are pointing out to those patients who visited the GP and were having a (first) ICPC-diagnosis of tinnitus. For example, if the patient would have visited the GP with a list of complaints in which tinnitus was not the most important and not scored by the GP, the patient will be classified as ‘patient without tinnitus’. This is very important to get clear throughout the text, including the abstract!

Line 134: ‘The presence of chronic diseases was dichotomized into two categories (zero and one or more). � from this it is unclear at which time this was assessed; at baseline of each patient or at the end of follow-up of each patient. Secondly, which domains of chronic diseases were considered? Mental diseases?

Results section;

Line 174; ‘prescription for tinnitus’; it is unclear to me how the authors can conclude on the disease related to the prescribed drugs and whether this is really related to the tinnitus (or to any other disease for which the patient was visiting the GP at the same time). If this is based on having a tinnitus diagnosis and having a prescription at the same data, then it is still to be debated if the prescription is to treat the tinnitus.

Line 199; ‘during tinnitus’; this is an assumption that people were (still) having tinnitus after setting the diagnosis (although logically, this is not clear )

Discussion:

Line 215: Our findings reveal that the number of patients contacting their GP for 215 tinnitus is low � this conclusion is false as the database does not contain data on people with tinnitus who didn’t attend the GP. At the same time I would mention the actual years of observation in this sentence so that the reader does understand in which time the ‘increase’ was observed.

Line 235-250: In the discussion the prevalence and incidence of tinnitus is discussed with explanations for higher numbers in older people and assumptions that this can be related to ‘mobile technology usage’ amongst others. However I do miss a more critical assessment of the rationale for the findings. For example, also ‘changes over years in behaviour of patients to seek help’ can attribute to increasing records of a tinnitus diagnosis. Then this should also appear for other diseases. As the authors are health care registration specialists I have no doubts that they are able to improve the quality of this manuscripts by elaborating on this before drawing conclusions that it is only a higher disease prevalence or incidence by itself as the only explanation for the presented findings. Apart from this, in current literature there is a lack of knowledge about which leisure context do contribute the most to noise induced hearing loss (including tinnitus). Now the authors state only mobile technology usage as main focus in the abstract, introduction and discussion, but also other leisure contexts can be the ‘main cause’ (e.g. bars, dancings, festivals).

Line 252: ‘Since guidelines for tinnitus are not widely implemented across Europe, management and treatment can vary widely across countries and GPs.’ � With a reference on the paper of Cima et al, which is a summery of European guidelines without any original quality assessment of guidelines and individual made recommendations, how can the authors conclude on ‘not widely implemented’? I do agree that treatment vary widely and for sure following guidelines is far from perfect in health care settings, but the first part of the sentence does not hold any basis.

Line 260: ‘Nasal corticosteroids are the initial therapy for sudden sensorineural hearing loss’ � Ok, this is a mistake. I will help you out; nasal corticosteroids are used for different diseases mostly rhinosinusitis (related) problems. The authors have a mix-up with ‘oral or intratympanic corticosteroids in which the first is the initial therapy for sudden hearing loss with intratympanic considered as salvage therapy. Please correct this and the rationale around the nasal corticosteroids findings in the manuscript.

Line 263-265: ‘. Although this is not the first line of treatment for tinnitus and evidence is 264 inconclusive, review studies widely describe that these may positively influence the subjective experience of tinnitus.31,37’ � this statement is not in line with the current evidence nor the opinion on its effectiveness. Please read Cochrane Reviews on these treatments or nationale guidelines on this; these drugs are not considered effective and are not part of any clinical guideline on tinnitus treatment. So, not a first line, but not any line treatment and suggesting ‘may a positive influence’ is misleading readers. Off course, I do agree that a guideline for tinnitus for GPs could be an important step to improve treatments standards and uniformization, and could limit referrals to second line of care such as discussed in the next paragraph, but the rationale for this should be correct.

Line 331; This could indicate that a large proportion of these 332 patients require medical attention other than their GP. � There are several reasons for the referral which are also mentioned in the discussion by the authors. It is not sure if the patient required medical attention as reason for referral. Maybe the GP was limited by facility or knowledge to provide care needed by the patient, or because they were just following guidelines (and as the authors stated; there are no GP guidelines..)

Reviewer #2: In general: well written article about the development of tinnitus in the Dutch General practice. The discussion session is very strong and I like the suggestions for future studies.

Some suggestions:

Title: This is a little bit long. Can you trim this?

r 45: tinnitus is not always chronic. I would prefer that you remove the word chronic in the sentence.

r 53 – 55: “Various treatments and care for tinnitus can be provided mainly to reduce 55 or cope with the complaints. For example, drug therapies used to treat tinnitus, like antihistamines, 56 antidepressants, or antiepileptics, may provide symptom relief but do not offer a complete cure for tinnitus”

Like you also mention in the discussion: there is no guideline of tinnitus that recommends drug therapy. I would recommend giving cognitive behavior therapy as an example. This is far more accepted.

R 69: “Due to the preventable nature of tinnitus” What do you mean? I think you refer to the hearing loss and noise exposure? However, hearing loss is just one of the influencing factors. There are much more influencing factors like stress, anxiety, depression, temporomandibular disorders.

Table 1:

No chronic disease: 5442, should that be 5,442?

6. PLOS authors have the option to publish the peer review history of their article (what does this mean?). If published, this will include your full peer review and any attached files.

Reviewer #1: **Yes: **A.L. (Diane) Smit

Reviewer #2: **Yes: **Annemarie van der Wal

---

## [Author Response · Author response to Decision Letter 0]

18 Sep 2024

Response letter - Revision PlosONE

Tinnitus; a growing public health problem, but what is the impact on Dutch general practices? A retrospective study using routine healthcare data

General

We would like to thank the reviewers and editor for the comments that helped improving the manuscript and the opportunity to resubmit the manuscript. Please find our point-by-point reply below.

Comments from the Journal editor:

Answer: The unmarked version of the revised manuscript (‘Manuscript’) has been updated to meet the PLOS ONE’s style requirements.

Answer: We did not receive a grant/ grant number for this specific study and therefore have not mentioned under ‘Funding information’. The Nivel Primary Care Database however is funded by the Dutch Ministry of Health, Welfare and Sports which we have mentioned under ‘Financial Disclosure’. These sections do not match, since 'Funding information' has no free text box.

3. We note that you have indicated that there are restrictions to data sharing for this study. PLOS only allows data to be available upon request if there are legal or ethical restrictions on sharing data publicly. For more information on unacceptable data access restrictions, please see http://journals.plos.org/plosone/s/data-availability - loc-unacceptable-data-access-restrictions. Before we proceed with your manuscript, please address the following prompts

b) If there are no restrictions, please upload the minimal anonymized data set necessary to replicate your study findings to a stable, public repository and provide us with the relevant URLs, DOIs, or accession numbers. For a list of recommended repositories, please see https://journals.plos.org/plosone/s/recommended-repositories. You also have the option of uploading the data as Supporting Information files, but we would recommend depositing data directly to a data repository if possible.

Answer: Data of the Nivel-PCD contains potentially identifying or sensitive patient information. Access to data in the Nivel Primary Care Database (Nivel-PCD) is subject to Nivel Primary Care Database governance codes. Requests for access to the data can be directed at gegevensaanvragen@nivel.nl Restrictions involve approval by the appropriate Nivel Primary Care Database governance bodies (privacy committee and steering committee).

Comments from the reviewers

Reviewer #1

In this manuscript the authors describe an interesting retrospective study to report the morbidity trends and healthcare utilization among patients with tinnitus at Dutch general practices (GP), and comparing overall healthcare utilization before and after tinnitus to similar patients without tinnitus using health care databases. It is a topic of high interest, with clear methods, high quality data and adequate description of results and its discussion. I strongly recommend studies on this for providing knowledge on health care consumption and its potential impact on applying health care resources to those in need. However, the manuscript misses quality based on flaws in details on tinnitus and rationale for statements quoted throughout the manuscript. I strongly advise to revise the manuscript on this which at this moment strongly limit the quality of the report and could easily be adjusted.

Author’s response: We thank the reviewer for the compliments and for the suggestions below. 

Introduction

1. Line 45; this is the definition of subjective tinnitus; please add this to the sentence.

Response 1: We agree that our definition refers to subjective tinnitus, the most common form of tinnitus. In most of the articles we refer to, tinnitus is generally discussed without distinguishing between subjective and objective forms. Since the majority of these articles use the term "tinnitus" in this way, we believe it would improve the readability of our article to follow this convention and avoid making a distinction between subjective and objective tinnitus.

2. Line 48: ‘ranging from 5% and 43%’; details are missing about these percentages, for example ‘of people from the adult general population’?. Besides this, it’s incorrect to state that with respect to these numbers they all ‘suffer from tinnitus’. About at least 25% of the people with tinnitus do not have any complaints from their tinnitus. Please be cautious with this phrasing on suffering (e.g. also in line 228)

Response 2: Both references of line 48, which are systematic reviews that estimated global prevalence of tinnitus, included or reported on 'adult population studies reporting the prevalence of tinnitus'. We have added this information to the percentages in line 48. In addition, we have replaced the word suffer with experience, as we agree that not all people with tinnitus are suffering from tinnitus. We have also amended suffer in experience in line 55.

3. Line 51; ‘by occupation and mobile technology usage, its prevalence is expected to rise (1.4% in each 5-year birth cohort). ‘ � this statement is speculative. So far there is no robust evidence on the rise of the prevalence nor on the etiology of this ‘expected rise’. So please refrain from speculations and describe this as an hypothesis, knowledge gap or topic of interest for future health care purposes. This is the same reason that also the title of the manuscript is somewhat misleading: ‘Tinnitus; a growing health problem….’. As the evidence base for this statement is absent and the current manuscript is on GP diagnosis we can’t make firm conclusions on this. Therefore I would suggest to rephrase this title slightly

Response 3: Thank you for your suggestion. We have eliminated the speculative statement and revised the sentence. In addition, we rephrased the title as follows: “The impact of tinnitus on Dutch general practices: a retrospective study using routine healthcare data"

4. Line 55: ‘For example, drug therapies used to treat tinnitus, like antihistamines, 56 antidepressants, or antiepileptics, may provide symptom relief but do not offer a complete cure for tinnitus. � this statement is not according to national treatment guidelines nor according to the current evidence base on the effectivity of the mentioned therapies.

Response 4: The Dutch guideline for tinnitus of the federation of medical specialists’ states "Do not prescribe medications for tinnitus. Comorbidity should be treated, possibly with medication, if necessary." We have improved line 53-55 as follows: “Various treatments and care for patients with tinnitus are available , mainly to treat its comorbidities. 10 There is no specific drug proven effective against tinnitus available.”

5. Line 65; ‘cross-sectional study by Rademaker et al. indicated that healthcare use is higher in 66 patients with tinnitus.’ please indicate the comparison which was made (in this case compared to those without a diagnosis of tinnitus).

Response 5: We have added the comparison that was made.

6. Line 96: ‘patients with tinnitus’; Although in the text it is mentioned that ICPC-diagnosis were obtained, from this, the reader can’t understand that if you refer to ‘patients with tinnitus’ you are pointing out to those patients who visited the GP and were having a (first) ICPC-diagnosis of tinnitus. For example, if the patient would have visited the GP with a list of complaints in which tinnitus was not the most important and not scored by the GP, the patient will be classified as ‘patient without tinnitus’. This is very important to get clear throughout the text, including the abstract!

Response 6: Even if a patient has tinnitus, they may not be mentioned to their GP, or it might indeed not be recorded by the GP. To address this nuance, we have added the term "recorded " before tinnitus at various points in our abstract (lines 21 and 23) and in the methods section (line 97), before it is explained how tinnitus is defined in this study.

7. Line 134: ‘The presence of chronic diseases was dichotomized into two categories (zero and one or more). � from this it is unclear at which time this was assessed; at baseline of each patient or at the end of follow-up of each patient. Secondly, which domains of chronic diseases were considered? Mental diseases?

Response 7: We have added the time of assessment, which was at time of tinnitus diagnosis or index date of matched controls, to the manuscript. Additionally, a list of 109 chronic conditions was compiled for the Nivel-PCD based on literature and discussion groups. We have updated the manuscript to specify that it involves 109 chronic diseases, and the referenced article provides a detailed list of these conditions (i.e., disabilities, congenital anomalies, malignant cancers, diabetes mellitus, hypertension, inflammatory arthritis, psoriatic disease, and dementia).

Results section;

8. Line 174; ‘prescription for tinnitus’; it is unclear to me how the authors can conclude on the disease related to the prescribed drugs and whether this is really related to the tinnitus (or to any other disease for which the patient was visiting the GP at the same time). If this is based on having a tinnitus diagnosis and having a prescription at the same data, then it is still to be debated if the prescription is to treat the tinnitus.

Response 8: In Nivel-PCD, prescriptions include an ICPC code provided by the GP, indicating the diagnosis for which the medication was prescribed. To clarify this, we have added the following to the methods section on line 130: "To calculate the annual prescription percentage for tinnitus, we counted the number of patients who received at least one prescription linked to an ICPC code for tinnitus after consulting their GP." 

9. Line 199; ‘during tinnitus’; this is an assumption that people were (still) having tinnitus after setting the diagnosis (although logically, this is not clear )

Response 9: We agree that the type of data we use only show that tinnitus was recorded at a specific moment and do not indicate whether the condition is ongoing. Therefore, we have revised 'during tinnitus' on lines 202, 207, and 208 to 'after first contact for tinnitus.'

Discussion:

10. Line 215: Our findings reveal that the number of patients contacting their GP for 215 tinnitus is low � this conclusion is false as the database does not contain data on people with tinnitus who didn’t attend the GP. At the same time I would mention the actual years of observation in this sentence so that the reader does understand in which time the ‘increase’ was observed.

Response 10: Since systematic reviews and questionnaire studies indicate that the prevalence and incidence of tinnitus are actually higher (as noted in the introduction on lines 47-49: “Worldwide, an estimated 740 million adults experience any tinnitus, with overall prevalence ranging from 5% to 43% of the adult population, depending on severity,” and in lines 229-230: “Survey studies have shown that the number of people suffering from tinnitus is greater than what our study found”), it is important to acknowledge that our reported incidence is low. We believe that simply stating that incidence has increased over the past decade would lack context. We do agree that it is important to add a timeframe to this sentence, therefore we have added this detail to line 219: “Our findings reveal that the number of patients contacting their GP for tinnitus is low but has increased over the past decade, especially among those aged 20 to 44 years.”

11. Line 235-250: In the discussion the prevalence and incidence of tinnitus is discussed with explanations for higher numbers in older people and assumptions that this can be related to ‘mobile technology usage’ amongst others. However I do miss a more critical assessment of the rationale for the findings. For example, also ‘changes over years in behaviour of patients to seek help’ can attribute to increasing records of a tinnitus diagnosis. Then this should also appear for other diseases. As the authors are health care registration specialists I have no doubts that they are able to improve the quality of this manuscripts by elaborating on this before drawing conclusions that it is only a higher disease prevalence or incidence by itself as the only explanation for the presented findings. Apart from this, in current literature there is a lack of knowledge about which leisure context do contribute the most to noise induced hearing loss (including tinnitus). Now the authors state only mobile technology usage as main focus in the abstract, introduction and discussion, but also other leisure contexts can be the ‘main cause’ (e.g. bars, dancings, festivals).

Response 11: Thank you for your insightful feedback. We have described in our discussion the increased media awareness for tinnitus, whereby we did not describe that this might contribute to more help-seeking behavior for this complaint. This is now more clearly described in lines 252-254: "Another possible explanation for the increase in prevalence might be the level of awareness of tinnitus, possibly due to media attention for tinnitus that raise awareness in the general population. This might increase help seeking behavior at GPs." We also acknowledge that loud noise exposure can occur in various leisure activities beyond mobile technology use. We have added more details to line 250: " In addition, exposure to loud noise during leisure time activities such as visiting concerts and nightclubs as well as mobile technology usage (e.g. cell phones and music players) " to further support our discussion. We have also added the leisure context to our introduction in line 54: “Multiple risk factors are associated with tinnitus, such as loud noise exposure (e.g., during leisure time activities) and hearing loss.”

12. Line 252: ‘Since guidelines for tinnitus are not widely implemented across Europe, management and treatment can vary widely across countries and GPs.’ � With a reference on the paper of Cima et al, which is a summery of European guidelines without any original quality assessment of guidelines and individual made recommendations, how can the authors conclude on ‘not widely implemented’? I do agree that treatment vary widely and for sure following guidelines is far from perfect in health care settings, but the first part of the sentence does not hold any basis.

Response 12: A survey study of Cima et al. concluded ‘Most European countries do not have national clinical guidelines for the management of tinnitus. Reflective of this, clinical practices in tinnitus healthcare vary dramatically across countries”. We had mistakenly cited a different study of Cima et al., so we have corrected the reference. Additionally, we revised line 264 for clarity “Given that the majority of European countries do not have national guidelines for tinnitus, management and treatment can vary widely across countries and GPs”

13. Line 260: ‘Nasal corticosteroids are the initial therapy for sudden sensorineural hearing loss’ � Ok, this is a mistake. I will help you out; nasal corticosteroids are used for different diseases mostly rhinosinusitis (related) problems. The

---

## [Editor Report · Decision Letter 1]

29 Oct 2024

The impact of tinnitus on Dutch general practices: a retrospective study using routine healthcare data

PONE-D-24-13714R1

Dear Dr. Bes,

We’re pleased to inform you that your manuscript has been judged scientifically suitable for publication and will be formally accepted for publication once it meets all outstanding technical requirements.

Kind regards,

Sarah Michiels

Academic Editor

PLOS ONE

---

## [Editor Report · Acceptance letter]

6 Nov 2024

PONE-D-24-13714R1 

PLOS ONE

Dear Dr. Bes, 

I'm pleased to inform you that your manuscript has been deemed suitable for publication in PLOS ONE. Congratulations! Your manuscript is now being handed over to our production team.

Kind regards, 

on behalf of

Prof. Sarah Michiels 

Academic Editor

PLOS ONE